# Abducted Child's Best Interests versus the Theoretical Child's Best Interests: Australia, New Zealand and the Pacific

Mark Henaghan *, Christian Poland and Clement Kong

Faculty of Law, University of Auckland, Auckland 1010, New Zealand
* Correspondence: mark.henaghan@auckland.ac.nz

**Abstract:** A recent trend can be seen in jurisprudence concerning the Hague Convention on the Civil Aspects of International Child Abduction, at least in the Australasia/Pacific region. Courts are now more mindful of the abducted child in particular and will investigate the true impacts of returning the child to determine what is in their best interests, particularly in cases of domestic violence. This is a departure from the long-standing emphasis on returning abducted children promptly to their country of habitual residence, after which the courts of that country will make the final decision, because it is generally in the best interests of children to deter child abduction. This article compares various jurisdictions' approaches with the lens of whether the courts are preferring the particular child over the 'theoretical' child.

**Keywords:** child settled exception; grave risk exception; child objection exception; human rights exception

## 1. Introduction

Once the applicant satisfies the jurisdiction requirements for the child's return under the Hague Convention on the Civil Aspects of International Child Abduction (hereafter the Convention), article 12 generally requires the court to order that the child return to their country of habitual residence. This recognises the 'theoretical' child and the presumption that it is generally in the best interests of children to return abducted children promptly, thereby deterring future abductions.

However, the respondent may satisfy one of the exceptions to this under the Convention. These exceptions allow for a greater focus on the individual interests of the particular child who has been abducted. Once these exceptions have been considered by the court, they are weighed against the Convention's purpose to protect children against the harm of child abduction before the court makes a final decision on whether or not to return the child.

This article discusses the shift in jurisprudence concerning the most litigated exceptions under the Convention from what is in the interests of children generally to what is in the interests of the particular child who has been wrongfully removed from their country of habitual residence. The article looks to the jurisprudence in the three countries in the Australasia/Pacific region that are signatories to the Convention: Australia, Aotearoa New Zealand[1] and Fiji (HCCH 2022). Australia implemented the Convention into domestic law in the Family Law (Child Abduction Convention) Regulations 1986 (hereafter Australia Regulations), and Fiji did the same in its Family Law Regulations 2005 (hereafter Fiji Regulations). New Zealand initially incorporated the Convention in the Guardianship Amendment Act 1991, which was subsequently replaced by subpart 4 of the Care of Children Act 2004. The article also discusses how non-signatory countries in

---

[1]  Aotearoa is the Māori name for New Zealand, literally meaning 'the long white cloud'. Māori are the indigenous people of Aotearoa.

the Pacific—Samoa, Tonga, Papua New Guinea and the Cook Islands—address international child abduction cases in light of the Convention principles and the particular child's best interests.

## 2. Child Settled Exception

This exception is satisfied when the child was removed more than one year before the application was made, and the respondent proves that the child is now settled in their new environment.

### 2.1. Australia

Australian courts have preferred not to put any particular gloss on the meaning of 'settled'.[2] It is given its ordinary meaning. However, several factors that a court can consider to determine if the child is now settled include whether the child appeared content in their current environment; the child's subjective views of their current circumstances and the weight to be given to those views; whether the respondent has 'established a stable physical and financial environment'; how embedded the child is in their current community and the 'nature and circumstance of each child which might impact upon an assessment of whether each child is settled'.[3]

Unlike New Zealand and Fiji, Australian courts do not have a residual discretion to return a child when the child settled exception is made out.[4] This puts the particular child as paramount over the 'theoretical' child under the Convention. The Australian appellate court noted that New Zealand's position was different but said that this was because of how each country has implemented the exceptions into domestic law.[5] New Zealand's statute puts the child settled exception on par with the other exceptions, whereas the Australian regulations separate the child settled exception from the rest.

### 2.2. New Zealand

Unlike Australia, New Zealand courts have discussed the meaning of 'settled'. The question of whether a child is now settled at the date of hearing 'involves a consideration of physical, emotional and social issues. Not only must a child be physically and emotionally 'settled' in the new environment, he or she must also be socially integrated'.[6]

As discussed above, New Zealand courts retain a residual discretion to return a child even when the exception is satisfied. The majority of the Supreme Court in *Secretary for Justice (New Zealand Central Authority) v H J* thought that this discretion should be exercised when 'the best interests of the particular child [are] outweighed by the interests of other children in Hague Convention terms [so that] to decline return would send the wrong message to potential abductors'.[7] The majority was concerned about situations where, for example, the abducting parent has concealed the child until they have become 'settled'.[8]

Unfortunately, this approach was misinterpreted in *Simpson v Hamilton*.[9] The Court of Appeal held that a 'significant change of circumstances' since the Family Court's decision two years earlier allowed it to refuse to return the child, even though the child settled exception had not been made out, so there was no statutory discretion to do so.[10] The Supreme Court agreed with this approach to reassess the exceptions in light of new evidence but clarified that the child settled exception *was* made out.[11] This justified the Court of

---

2　*Director General, Dept of Community Services v M and C* [1998] FamCA 1518, (1998) 24 Fam LR 178 [52], [91].

3　*Department of Family and Community Services v Raho* [2013] FamCA 530 [244].

4　*Secretary, Department of Family and Community Services v Magoulas* [2018] FamCAFC 165, (2018) 61 Fam LR 117.

5　Ibid, [33].

6　*Secretary for Justice (New Zealand Central Authority) v H J* [2006] NZSC 97, [2007] 2 NZLR 289 [55]–[57].

7　Ibid, [50], [85].

8　Ibid, [87].

9　*Simpson v Hamilton* [2019] NZCA 579, [2019] NZFLR 338.

10　Ibid, [78]. See (Henaghan and Poland 2021, pp. 365–70).

11　*Simpson v Hamilton* [2020] NZSC 42, [2020] NZFLR 37 [45].

Appeal's ability to refuse return but rewrote its judgment in the process. It clearly thought that none of the exceptions were made out.[12]

The Court of Appeal has since opposed the 'balancing' approach taken by the majority of the Supreme Court in *Secretary for Justice v H J*.[13] Instead, the Court of Appeal thought that the discretion should be exercised in the best interests of the particular child, which was Elias CJ's dissenting view. However, the Court of Appeal made its comments during a case that raised the 'grave risk' exception, not the 'child settled' exception, so it was not bound to follow the Supreme Court's approach.

*2.3. Fiji*

Wati J in *PSJ v TR* determined that the child settled exception was made out and refused to return the children.[14] When they came to Fiji, they were two and three years old—but they were eight and nine years old at the time of the return application.[15] A welfare report was sought, which found that the children were settled in their school environment and that it would be detrimental to remove them from their mother, close relatives and wider community.[16] On the other hand, the children had to change schools regularly because of their mother's occupation. Wati J dismissed this concern:

'Initially to settle in a place every parent finds it difficult and there has to be some changes in living places and schools of the children. That does not mean that there is substantial instability in children's lives because of that.'[17]

Once the exception was made out, the Fijian court followed the New Zealand approach, rather than the Australian approach, by determining that the courts do have a residual discretion to consider whether returning the child is appropriate or not.[18] This is despite the identical structure and substance of Australia and Fiji's respective regulations.[19] Regulation 73(6) of the Fiji Regulations allows a court to make a return order even if any of the exceptions in reg 73(4) are met, but reg 73(4) does not include the child settled exception—which is instead in reg 73(2).[20] The Fijian courts do not seem to have justified why they have reached a different position than their Australian counterparts. The practical effect is that the particular child's best interests must be weighed against the 'theoretical' child and their best interests.

## 3. Grave Risk Exception

Alternatively, the respondent can prove there is a grave risk that returning the child would expose them to physical or psychological harm or otherwise place them in an intolerable situation. This exception requires courts to predict what may happen if the child is returned, based on the evidence.[21] That prediction does not need to be certain—but it does require, as the Australian courts have put it, clear and compelling evidence of a real risk of exposure to harm.[22] 'Grave risk' is to be given its ordinary meaning, rather than any narrow or broad construction.[23]

Returning a child will always involve some degree of disruption and anxiety, but the grave risk exception contemplates more than that.[24] However, if anyone returning

---

12   *Simpson v Hamilton* (CA), note 9, [64].
13   *LRR v COL* [2020] NZCA 209, [2020] 2 NZLR 610 [99].
14   *PSJ v TR* [2015] FJHCFD 3 [70].
15   Ibid, [65].
16   Ibid, [66].
17   Ibid, [68].
18   Ibid, [58]; *PSJ v Lal* [2020] FJHCFD 6 [69].
19   Family Law (Child Abduction Convention) Regulations 1986 (Aus), regs 16(2)–16(3); Family Law Regulations 2005 (Fiji), regs 73(3)–73(4).
20   The Australian court applied this logic to Australia's equivalent provisions in *Magoulas*, note 4, [18].
21   *DP v Commonwealth Central Authority* [2001] HCA 39, (2001) 206 CLR 401 [41]; *LRR v COL*, note 13, [90].
22   *DP*, note 21, [43]; *LRR v COL*, note 13, [90].
23   *DP*, note 21, [44]; *LRR v COL*, note 13, [87].
24   *DP*, note 21, [45].

to that country would face a grave risk of harm (like warfare or civil unrest), then that is sufficient.[25] The welfare of the child is not the paramount consideration, and instead, Australian courts have discussed how the 'intention' of the Convention is to severely limit the courts of the country where the child has been taken.[26]

Australia and New Zealand have taken a similar path regarding the grave risk exception. Historically, the courts would tend to trust the overseas court to resolve disputes and best protect the child.[27] The Family Court of Australia, for instance, commented:

'There is no reason why this court should not assume that once the child is so returned, the courts in that country are not appropriately equipped to make suitable arrangements for the child's welfare.'[28]

Nowadays, though, both jurisdictions place greater weight on the impact on the child of the return. Although the HCCH's *Guide to Good Practice* on art 13(1)(b), published in 2020, is aimed at guiding courts, practitioners and Central Authorities on how to strike the right balance between the particular child and the 'theoretical' child, it has not been cited frequently by the courts.[29]

This section of the article canvasses the recent developments in Australia, New Zealand and Fiji regarding the grave risk exception.

### 3.1. Australia

The 2020 decision of *Walpole v Secretary, Department of Communities and Justice* reflects a greater understanding of domestic violence and how the primary victim being in danger can constitute a grave risk to their child.[30] The mother removed the children from New Zealand to Australia after suffering violence from the father and fearing for her life. As a victim of intimate partner violence, she would struggle to escape the abusive cycle if the children were ordered to return to New Zealand and she accompanied them. This was an intolerable situation for the children, as were poverty and poor living conditions.[31] Although 'New Zealand has sophisticated systems in place to protect victims of family violence', the father continued to inflict violence despite protection orders and imprisonment.[32] This meant the children could not be returned, which rightfully placed the children's best interests above the Convention's general principle to return children promptly.

The trend seen in *Walpole* was continued in December 2022 with the Family Law (Child Abduction Convention) Amendment (Family Violence) Regulations 2022. This amended the Australia Regulations to 'provide additional safeguards to parents and children fleeing family and domestic violence' (Dreyfus 2022). The amendment clarifies that when the court is considering if the grave risk exception is made out, the court may—'regardless of whether the court is satisfied that family violence has occurred, will occur or is likely to occur'—have regard to:

(a)　any risk that returning the child would result in them being subject to, or exposed to, family violence'; and

(b)　the extent to which the child could be protected from such risk.[33]

---

[25]　*Genish-Grant v Director-General, Department of Community Services* [2002] FamCA 346, (2002) 29 Fam LR 51 [20].

[26]　*Director-General of Family and Community Services v Davis* (1990) 14 Fam LR 381, FamCAFC, pp. 383–84.

[27]　See *A v Central Authority for New Zealand* [1996] 2 NZLR 517, CA, p. 522.

[28]　*Gsponer v Johnstone* (1988) 12 Fam LR 755, FamCAFC, p. 768. See also *Murray v Tam, Director, Family Services* (1993) 16 Fam LR 982, FamCAFC, pp. 1001–02.

[29]　Our study of the case-law identified only a handful of New Zealand judgments in which the *Guide to Good Practice* has been discussed: *LRR v COL*, note 13, [103]; *Roberts v Cresswell* [2022] NZHC 2337 [59]; *Creek v Hodder* [2022] NZFC 11049 [12]–[13]; *Parish v McDonald* [2022] NZHC 3022 [50], [55]. To our knowledge, the *Guide to Good Practice* has not been discussed by the Australian or Fijian courts. We hope that courts begin to appreciate the helpful guidance in this document and incorporate it into the court's reasoning.

[30]　*Walpole v Secretary, Department of Communities and Justice* [2020] FamCAFC 65, (2020) 60 Fam LR 409.

[31]　Ibid, [73].

[32]　Ibid, [75].

[33]　Family Law (Child Abduction Convention) Regulations 1986 (Aus), reg 16(3) Note 1. "Family violence" is defined in Family Law Act 1975 (Cth), s 4AB(1) to mean 'violent, threatening or other behaviour by a person

If the court is not convinced that the grave risk exception is made out, the court can still impose conditions on return orders regardless of whether it is satisfied that the risks will, or are likely to, eventuate, and regardless of whether the risk has eventuated in the past or not.[34] When proposing that a condition be imposed, the court may take into account its proportionality, reasonable practicability of compliance, enforceability and whether it 'would usurp the regular functions of the courts or authorities in the child's state of habitual residence', as well as any other matters it thinks relevant.[35]

However, the quid pro quo of the amending regulations is that if that court is considering whether to refuse to return the child, and a party or the child's lawyer raises a possible condition that could be included in the return order to reduce the risks faced by returning the child, the court must consider whether it would be appropriate to impose that condition.[36] The court may also consider any other measures reasonably likely to reduce the risks, as well as any other matters it thinks relevant.[37]

These new Regulations are commendable for signalling to the courts that domestic violence is indeed a relevant and important consideration under the grave risk exception. It is disappointing, however, that they say the court *may* have regard to the risk that returning the child would result in them being subjected or exposed to family violence, rather than saying that the court *must* take into account such a risk.

*3.2. New Zealand*

3.2.1. LRR v COL

The New Zealand case of *LRR v COL*, decided by the Court of Appeal, was similar to *Walpole*: both courts refused to return the child because of domestic violence concerns, and new evidence was crucial to both decisions. Returning the child in *LRR v COL* would create an intolerable situation because his mother had no other viable options except returning to the father's violence. She would struggle financially, her frail mental health and suicidal thoughts would probably relapse and her parenting capacity may have been impaired. It was therefore shown that there was a grave risk of harm and an intolerable situation if the child was returned.

While the Court of Appeal thought it was not taking a new direction, the proceeding became an extensive inquiry into the welfare of the child to determine whether the exception was made out.[38] The Court specifically held that the paramountcy principle *does* apply to Hague Convention proceedings in New Zealand, which is new.[39] This clarifies that if the applicant satisfies an exception under the Convention, they have sufficiently displaced the general presumption that a prompt return is in the child's best interests. Therefore, at the discretion stage, the courts should now consider the child's welfare and best interests as the *paramount* consideration, rather than weighing them against the purposes of the Convention (Henaghan and Poland 2021, pp. 373–74). The child should no longer be punished because of 'countervailing policy objectives pertaining to general deterrence of child abduction worldwide' (Murphy 2020, p. 43). Australian courts, however, have rejected this argument that the child's interests can (and should) be paramount in what are ultimately questions of forum rather than the substantive proceeding.[40]

The Court of Appeal also emphasised that courts need to inquire into whether protective measures by the country of habitual residence will actually work to mitigate the

---

grave risk, rather than assume they will.[41] Finally, it would be 'inconceivable' to order the child's return regardless once the grave risk exception is made out—whereas the Australian Family Court in *Walpole* still inquired into whether it should exercise its discretion or not.[42]

3.2.2. Roberts v Cresswell

*LRR v COL*'s powerful emphasis of the particular child and their best interests has since been walked back somewhat by the Court of Appeal in *Roberts v Cresswell*.

The case had a turbulent path in the appellate courts. Doogue J in the High Court quashed a Family Court order to return two children to France. Her Honour recognised the importance of the Court of Appeal's change of approach in *LRR v COL*.[43] Courts have previously been unduly narrow when approaching the affirmative defences, when the reality (in her view) is that the Convention puts the best interests of the children at the forefront.[44] The defences are therefore just as important to the Convention's effective operation as the jurisdictional grounds.[45] Hague Convention cases must now be viewed in this different light post-*LRR v COL*.[46]

Applying this approach to the facts, Doogue J found, with the benefit of expert evidence, that the children would find it distressing to be separated from their mother.[47] The Family Court Judge placed insufficient weight on this fact and wrongly relied on protective measures that *may* be put in place by the father to ensure the children remained in the mother's care on return to France pending the substantive decision.[48] The father had intense business commitments that meant that the children would be in an intolerable situation where they were not in either their mother's or father's care, given the father's unwillingness for the mother to care for the children pending the substantive decision.[49]

These matters could have been resolved with Court orders without resorting to the affirmative grave risk defence.[50] However, the mother provided further evidence that she had been abused by the father and resultingly suffered from PTSD, which would be triggered if she had to return to France with the children. Doogue J found these violent incidents were highly plausible based on the evidence, and, with the benefit of a doctor's report, that the mother's mental health deteriorated in France, and this would likely be triggered upon return. The mother had few employment prospects in France, given that she was not fluent in French and she had previously only worked in the father's business.[51] She would receive State support 'at the lower end of a standard of living index', and she had no social network in France.[52]

Her Honour concluded that the Family Court Judge did not apply *LRR v COL* correctly, overlooked psychiatrist evidence and instead found that the mother was a good parent in both France and New Zealand.[53] There was therefore a grave risk of placing the children in an intolerable situation if they were not in their mother's care as the primary parent, as well as the consequences that would come for the children from their mother's PTSD likely being triggered and her parenting being impaired. Of course, once the grave risk exception is made out, 'it is impossible to conceive of circumstances in which . . . it would be a legitimate exercise of the discretion nevertheless to order the child's return'.[54]

---

41   *LRR v COL*, note 13, [113]–[114].
42   Ibid, [96], [119]; *Walpole*, note 30, [74]–[77].
43   *Cresswell v Roberts* [2022] NZHC 1265 [65].
44   Ibid, [62].
45   Ibid, [60].
46   Ibid, [67].
47   Ibid, [108]
48   Ibid, [107].
49   Ibid, [102].
50   Ibid, [124].
51   Ibid, [194].
52   Ibid, [195]–[196].
53   Ibid, [202].
54   Ibid, [57].

However, the Court of Appeal set aside Doogue J's ruling and reinstated the Family Court order to return two children to France.[55] The father was permitted by the Court to adduce updated evidence, including evidence from a clinical psychologist that reviewed the mother's psychiatric evidence that convinced the High Court that a return to France would trigger the mother's PTSD. Goddard J, writing for the Court, accepted the father's argument that the mother's expert's evidence was too heavily relied upon by the lower court. It did not set out the criteria it used for the diagnosis, and it lacked the balance necessary for expert evidence that is meant to assist the court rather than advocate for one party.[56] It was the mother's lawyers' responsibility to ensure that the evidence was suitable for use in legal proceedings, rather than the expert.[57] The mother had also since come to contemplate returning with the children to France, and that material change in circumstances was relevant for what the conditions of return would be and how tolerable they will be for the children.[58] The father applied for a modification that would ensure that the children would not be separated from the mother (their primary carer) for a prolonged period.[59]

Goddard J described *LRR v COL* as a 'deliberate shift in emphasis'.[60] This meant the mother's family violence assertions and her psychological well-being upon return were both relevant factors not to be discounted.[61] However, given the material change in circumstances on appeal, return was far more tolerable—she would not live with the father, she and the children would not be exposed to physical violence and the risk of psychological violence could be controlled.[62] The Court refused to go into the work opportunities available to the mother in France, but there were options such as remote work, and the father agreed to financially support her and there were certain welfare entitlements available. There were counselling and mental health services available, and the French Family Court could provide further protective measures if needed—which the father would very likely comply with, unlike the father's history of non-compliance in *LRR v COL*. Simply put, a return to France would result in significant stress for the mother, which would have 'some adverse effects' for the children, but that was not a grave risk of an intolerable situation for them. She remained an effective and competent parent when living with the father.

The Supreme Court declined leave to appeal.[63] The Supreme Court found that the proposed appeal would only challenge the Court of Appeal's assessment of the facts, rather than how the approach in *LRR v COL* was applied to those facts.[64] However, the Supreme Court indicated that it may be open to hearing a future appeal:

'The high point of the applicant's proposed appeal is that the reforms reflected in *LRR v COL* are difficult and there is a need in some respects for further exposition of the relevant standard. The only one of the issues raised in this case that, in our view, may raise a question of general or public importance is that relating to the need for a wider understanding of domestic abuse, including recognition of the role of financial disparity, inequality of arms and legal processes in such abuse.'[65]

*Roberts v Cresswell*, therefore, stands for the proposition that respondents must satisfy that the alleged grave risk does actually exist before the particular child's best interests become compelling. Where that risk hinges on a psychiatric diagnosis, parties must be careful that their experts present their opinions in a cogent and comprehensive manner. It

---

55　*Roberts v Cresswell* [2023] NZCA 36 [150]–[151].
56　Ibid, [150]–[151].
57　Ibid, [152].
58　Ibid, [164].
59　Ibid, [167].
60　Ibid, [192].
61　Ibid, [194].
62　Ibid, [195].
63　*Cresswell v Roberts* [2023] NZSC 62 [15].
64　Ibid, [15].
65　Ibid, [14].

would be preferable for New Zealand courts to appoint one expert (or encourage parties to agree to a joint expert) as part of good case management practice (HCCH 2020, [90]).

### 3.3. Fiji

In *PSJ v Lal*, Wati J utilised the same principles as Australia and New Zealand.[66] The onus of proving the exception is on the respondent, and there is a 'very heavy' burden of proof.[67] The harm must be 'severe and substantial', and the risk must be much higher than merely 'unacceptable'.[68]

Her Honour found that the grave risk exception was not met. Although the father had a protection order against him, his supervised contact with the child was positive and there was no evidence to suggest he was inherently violent.[69] Wati J also thought that conditions could be imposed to ensure the child's safety upon return.[70] This decision, therefore, reflects the trend seen in Australia and New Zealand to treat domestic violence as a serious and relevant consideration for the grave risk defence but also to recognise the gravity of the risk required and that the risk may be properly mitigated. This matches the *Guide to Good Practice*, which requires courts to assess whether the effect on the child 'meets the high threshold of the grave risk exception, taking into account the availability of protective measures to address the grave risk' (HCCH 2020, [64]).

## 4. Child Objection Exception

Under Article 13 of the Convention, a court may refuse to order the return of the child if it finds that the child objects to being returned and has attained an age and degree of maturity at which it is appropriate to take account of their views. All three jurisdictions have added another requirement to this exception: that 'the child's objection shows a strength of feeling beyond the mere expression of a preference or of ordinary wishes'.[71] Australia and Fiji have added this in their domestic regulations, while New Zealand has imported this requirement through case law.[72]

In New Zealand, section 106(1)(d) of the Care of Children Act 2004 phrases the exception differently. Instead of whether the child's views should be taken into account, the question is what weight should be given to their views. However, this does not make a practical difference compared to Australia and Fiji because all three countries agree on the 'shades of grey' approach taken by Balcombe LJ in *Re R (Child Abduction: Acquiescence)*.[73] This approach assigns different levels of weight to each child's objection, rather than Millett LJ's 'in or out' approach: if a child is of sufficient age and degree of maturity then the court usually must not return the child against their wishes, but if the child is not of sufficient age or maturity then the court normally must return the child.[74]

All three jurisdictions have developed similar principles for the child objection exception. For instance, the objection must be to returning to the originating country, not to a particular parent.[75] A child's objection must be valid, reasonable, freely held and stronger than a mere preference.[76] The weight that is assigned depends on all the surrounding circumstances, including the child's age and maturity, as well as the rationality, cogency, strength and independence of their views.[77] Judges have thought that ten-year-olds or

---

66  *PSJ v Lal*, note 18.
67  Ibid, [72].
68  Ibid, [73].
69  Ibid, [81].
70  Ibid, [85].
71  Family Law (Child Abduction Convention) Regulations 1986 (Aus), reg 16(3)(c)(ii).
72  *S v M* [1993] NZFLR 584 (FC) 591; *Karly v Karly* [2017] NZFC 10030 [52].
73  *Re R (Child Abduction: Acquiescence)* [1995] 1 FLR 716, CA.
74  *De L*, note 40, 656; *White v Northumberland* [2006] NZFLR 1105, CA, [38]; *PSJ v VK* [2018] FJHCFD 1 [84].
75  *De L*, note 40, 655; *Karly*, note 72, [52]; *PSJ v VK*, note 74, [84].
76  *S v M* [1993] NZFLR 584 (FC) 591; *Karly*, note 72, [52].
77  *RCB v Forrest* [2012] HCA 47, (2012) 247 CLR 304; *S v S* [1999] 3 NZLR 513, HC, pp. 522–23; *Robinson v Robinson* [2020] NZHC 1765 [83]–[85]; *PSJ v VK*, note 74, [84]–[86].

younger are usually too young to assign their views any weight, but there are examples where considerable weight has been attached to the views of eight- and nine-year-olds (Caldwell 2008, pp. 85–86). Usually little to no weight is given to a child's views that have been influenced by the abducting parent.[78] Finally, when exercising the residual discretion, 'the court must balance the nature and strength of the child's objections against 'the Convention considerations' like comity and deterring future abductions, also known as the 'theoretical' child.[79]

For example, the child in *PSJ v VK* said she grew an attachment to her family in Fiji.[80] She did not enjoy living in New Zealand, as she was bullied there and suffered physical abuse and neglect from her mother.[81] Despite only being seven years old, the courts acknowledged that she had a sufficient degree of maturity to freely recognise what was in her best interests.[82] Wati J ordered that she should not be returned.[83]

## 5. Human Rights Exception

The human rights exception, under article 20 of the Convention, applies when the child's return is not permitted by the fundamental principles of the requested state relating to the protection of human rights and fundamental freedoms. However, it is not litigated often and can be misunderstood (Davies 2013). For instance, the New Zealand Family Court held that Fiji's military coup and resulting restrictions on movement and expression did not satisfy this exception.[84] It thought that more harm was required, which conflates the exception with the grave risk exception and misunderstands that '[t]he breach of a right is itself a harm' (Davies 2013, p. 237).

In *Peterson v Piripi*, the mother argued that if her tamariki Māori (Māori child) was returned to Australia, the child would no longer be surrounded and absorbed by the whānau hapū (sub-tribe), culture, whenua (lands), language and her birthright.[85] The mother was of the view that the Hague Convention was trying to usurp hapū tikanga and undermining the hapū's right to self-determination, which was never ceded. Tikanga, the customary laws of the Indigenous Māori people of Aotearoa, is the first law of Aotearoa New Zealand and remains a part of the country's common law.[86]

Judge Howard-Sager considered this argument within the context of the human rights exception to the Hague Convention. In this case, the child had many maternal whānau (family) members in Australia who could help maintain the child's connection with their culture until the Australian courts decide on the substantive care and contact issues. Indeed, her whānau had whanaungatanga (kinship) obligations to the child to ensure she remains engaged with her culture. This meant that the child's right to engage with her cultural heritage should not be impacted by return.[87]

As for the argument regarding hapū sovereignty and self-determination, Judge Howard-Sager was clear that the case was concerned with the child, not the hapū. It was not proven that the child was able to whakapapa (prove ancestral ties) to the hapū, nor that the child had a whāngai (customary adoption) link to the hapū.[88] Therefore, the Judge was not satisfied that the hapū's tikanga was applicable to the child.[89] This

---

[78] *Director-General, Department of Child Safety v Milson* [2008] FamCA 872 [90]; *Robinson*, note 77, [84]; *PSJ v VK*, note 74, [84]–[86].
[79] *Milson*, note 78, [88]–[89].
[80] *PSJ v VK*, note 74.
[81] Ibid, [95].
[82] Ibid, [91].
[83] Ibid, [108].
[84] *APN v TMH [Child abduction: grave risk and human rights]* [2010] NZFLR 463, FC.
[85] *Peterson v Piripi* [2023] NZFC 2584 [126].
[86] *Ellis v R (Continuance)* [2022] NZSC 114, [2022] 1 NZLR 239 [18]–[23].
[87] *Peterson v Piripi*, note 85, [139].
[88] Ibid, [141]–[148].
[89] Ibid, [149].

meant there could be no argument that returning the child would usurp the hapū's right to self-determination and breach their tikanga. The child was ordered to return to Australia.

The case provides a novel argument concerning the interplay between tikanga Māori and the human rights exception under the Hague Convention. The particular child and their particular situation and best interests demanded that they not be returned because this would be contrary to their hapū's sovereignty, tikanga and cultural heritage. The Convention, however, required that the child be returned because none of the Convention exceptions could be satisfied to the evidentiary levels required by Western courts.

This raises a sensitive issue about filing whakapapa evidence, which is a taonga and may be considered tapū (cultural restriction).[90] It is understandable that the mother would not want to file such evidence. If such evidence was filed and accepted, it seems that the Judge may have considered that returning the child would be a breach of applicable tikanga rights and freedoms (such as sovereignty and self-determination) and so the defence would have been made out.[91] However, there were also tikanga obligations (such as whanaungatanga) on the child's family to ensure the child received cultural guidance while in Australia. These obligations may have made the evidence redundant and led to the same conclusion: that returning the child to Australia would not breach the child's human rights and fundamental freedoms.

## 6. Approaches from Some Non-Hague Convention Pacific Countries

Fiji is the only jurisdiction in the Pacific region that has signed the Convention. However, the courts of other Pacific countries have outlined the process they take for alleged child abductions. It is interesting to analyse how each jurisdiction balances a particular child against more general and theoretical concerns with international child abduction.

### 6.1. Samoa

*Wagner v Radke* clarified Samoa's approach to international child abduction.[92] First, Sapolu CJ outlined that it is appropriate to apply the principles and policy of the Convention and have regard for the common law. The principles were incorporated as a matter of customary international law, which automatically forms part of Samoa's domestic law even if Samoa was not a signatory to the Convention.

At common law, the question is whether it is more appropriate for the court of habitual residence to determine the matter, or if the current court is the appropriate forum to consider orders other than returning the child immediately.[93] Crucially, the welfare of the child is the *paramount* consideration. Under the Convention, the general rule is that abducted children are returned promptly to their country of habitual residence. This does not necessarily align with the welfare and best interests of the child.

Nonetheless, Sapolu CJ's mixed approach first allowed him to consider whether Samoa was the appropriate forum. However, all the relevant witnesses were overseas, and the family's permits to remain in Samoa had expired so any custody order from the Samoan courts would be short-lived. The Convention encourages the child's prompt return, and none of the exceptions were satisfied. Ultimately, it was in the welfare and best interests of the child for the German courts to determine custody.

The Samoa Family Law Commission is currently undertaking a review of family law. The Ombudsman and National Human Rights Institution have recommended that Samoa become a signatory to the Convention. This would 'reinforce and solidify the bearing of such international law locally'.[94]

---

90　Ibid, [142].
91　Ibid, [152].
92　*Wagner v Radke* [1997] WSSC 6, (2005) 1 PHRLD 67.
93　Ibid, referring to *Re L (Minors) (Wardship: Jurisdiction)* [1974] 1 WLR 250, CA; *Re F (A Minor) (Abduction: Custody Rights)* [1991] Fam 25, CA; and *Re A (Minors) (Abduction: Custody Rights)* [1992] Fam 106, CA.
94　Office of the Ombudsman/National Human Rights Institution Samoa, *Submission on the Review of Family Laws of Samoa* (9 February 2021) 4.

### 6.2. Tonga

Another approach was taken by the Supreme Court of Tonga in *Gorce v Miller*.[95] Unlike Samoa, the Court reinforced Tonga's intention to not sign the Convention. Instead, the Supreme Court of Tonga applied English case law decided before 1985 (as *Wagner v Radke* discussed). The Court did not apply case law from after 1985 because the United Kingdom had ratified the Convention from then on, so case law from after 1985 would embody principles that Tonga had not subscribed to itself. Although the Convention and its general principle to promptly return the child was not applicable, it was in the child's best interests for them to be returned.

### 6.3. Papua New Guinea

Instead of adopting the common law or Convention's approach, the courts of Papua New Guinea utilised its own legislation to justify returning an abducted child. Charmain Backhouse was successful in her child being returned to Australia using the Lukautim Pikinini Act 2015, which has the purpose to promote and protect the welfare of the child (Fox 2020).[96] This is the only instance where this statute has been used in a child abduction case. The Act has a broader focus on the best interests of that particular child generally with no particular guidelines regarding child abduction, which has left the courts with wide discretion.

### 6.4. Cook Islands

Lastly, in *Marsters v Richards*, the Cook Islands High Court did not conclusively decide if the Convention principles were applicable or not.[97] Either way, returning the child was in their welfare and best interests, which is the paramount consideration under the Infants Act 1908 (NZ). The general presumption is to return the child, as this is usually in the child's best interests. The courts should take international law into account, and it was possible that the Convention formed a principle of customary international law against child abduction.

## 7. Conclusions

The law on international child abduction in the Australasia/Pacific region is constantly evolving, and a consistent pattern can be seen where courts are more concerned with the particular child involved in the proceedings.

When exercising residual discretion once an exception to return is made out, New Zealand courts have not clarified what weight should be given to the child's welfare and best interests. The New Zealand Supreme Court in *Simpson v Hamilton* preferred to balance the general concerns of upholding the Convention against the welfare and best interests of the particular child, and on the facts, it was in the child's best interests for the child to remain in New Zealand. However, the Court of Appeal in *LRR v COL* thought that the paramountcy principle applies to Convention proceedings. The *LRR v COL* case was in the context of proven allegations of violence, where courts are less reluctant to move away from the purposes of the Convention. This issue remains open for debate, as although the Supreme Court declined leave to appeal in *Roberts v Cresswell*, it signalled that it was open to hearing future cases in this area.

A key area for the Convention in the future is the grave risk exception, especially as it is the most litigated and successfully used exception, accounting for 25 percent of judicial refusals globally (Lowe and Stephens 2018), and because of the HCCH's development of the *Guide to Good Practice* on the grave risk exception.

Historically, courts tended to defer to the overseas court to make the final decision. This meant that the child was normally returned because it is in the best interests of children

---

[95]   *Gorce v Miller* [2003] TOSC 46, (2005) 1 PHRLD 8.

[96]   Lukautim Pikinini Act 2015, s 5(1).

[97]   *Marsters v Richards* DP 4/2008, (2011) 3 PHRLD 8.

generally to deter child abduction. Now, cases like *Walpole* in Australia and *LRR v COL* in New Zealand showcase a fresh approach that investigates the true impacts of returning the child to a grave risk or intolerable situation. This aligns with the *Guide to Good Practice*'s advice in cases involving domestic violence allegations to focus on 'the effect of domestic violence on the child upon his or her return' (HCCH 2020, [58]). However, cases such as *Roberts v Cresswell* in New Zealand and *PSJ v Lal* in Fiji remind us that 'courts should consider the availability, adequacy and effectiveness of measures protecting the child from the grave risk' (ibid., [59]).

The welfare and best interests of the particular child have entered the analysis, which is a positive development because the particular child should not suffer because of the 'theoretical' child (Murphy 2020, p. 43). The Convention's role of deterring child abduction is less overriding when it comes to ensuring the best interests of the particular children involved (ibid.). This is also the approach taken by all four non-signatory Pacific countries discussed in this article. They have the child's welfare and best interests at the heart of the inquiry because they are not bound to follow Convention procedures. However, they still recognise that child abduction is wrong and that children should be returned where practicable.

Cases involving grave risks to children and intolerable situations, particularly where the child's primary caregiver experienced violence, are now rightly given individual attention to ensure that each particular child who faces these circumstances does not suffer any harm. As a general rule, if the children are suffering a risk of harm, they are not returned because their individual interests are given more weight than the traditional policy of the Convention, which is to return children in most cases on the assumption that the authorities in their country will protect them.

This article has shown that it is inevitable that the wording and thrust of the Convention will be interpreted differently, even between countries like Australia and New Zealand that have much in common yet still have different approaches. The approaches are likely to be even more diverse between countries of different histories or cultures. The article also highlights the need for the Convention to remain fit for purpose, with the practice continuing to evolve to reflect changing circumstances and the nature of abductions. For example, children's views have emerged as an essential part of family law decision-making, but the child objection exception has been interpreted narrowly and does not reflect current thinking on child participation.

Finally, non-signatory states, such as the Pacific countries mentioned in this article, generally follow the Convention principles. Efforts should continue to be made to encourage and support these countries to sign on to the Convention, although it could be argued that the Convention is outdated and unnecessary given modern advances in technology. The courts in the country of habitual residence could instead conduct a remote hearing to determine the care and contact issue while the child remains overseas. This would prevent proceedings from dragging on, with potentially multiple courts hearing the Convention case and then multiple courts hearing the substantive case regarding care and contact. It would also satisfy the often-touted 'purposes of the Convention'—such as deterring future child abductions and preventing forum-shopping—because the case would always be heard by the most appropriate forum, namely the courts of the country of habitual residence. The interests of the particular child would govern the outcome of the dispute from the get-go.

**Author Contributions:** Conceptualization, M.H.; Methodology, M.H.; Investigation, C.P. and C.K.; Writing—original draft, M.H., C.P. and C.K.; Writing—review & editing, C.P.; Supervision, M.H.; Project administration, M.H. All authors have read and agreed to the published version of the manuscript.

**Funding:** This research received no external funding.

**Institutional Review Board Statement:** Not applicable.

**Informed Consent Statement:** Not applicable.

**Data Availability Statement:** Not applicable.

**Conflicts of Interest:** The authors declare no conflict of interest.

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
