# Peer review of "Abducted Child’s Best Interests versus the Theoretical Child’s Best Interests: Australia, New Zealand and the Pacific"

_laws, 2022_

Round 1
Author Response
We thank the reviewer for their thoughtful and valuable comments. We have adopted all their comments, with responses below:
- We have incorporated the 2022 Australian amendment regulations and the HCCH Guide to Good Practice, and provided our thoughts upon both.
- Line 83: We have clarified that the reference is to the majority decision in H J.
- Line 122-132: We have restructured this section to avoid any confusion.
- Line 133-144: We have incorporated the 2022 Australian amendment regulations to justify the Walpole case being part of a trend in the Australia jurisdiction.
- Line 147-162: We have clarified both here and in the conclusion.
- Line 156: We have clarified this.
- Line 162: We have clarified this and provided support that the 1996 case remains the leading case on this point.
- Line 192: We have clarified this, although it would not be correct to have the phrase at the beginning of the sentence.
- Line 250-251: We have added to this section.
- Line 263: We have amended this to avoid confusion.
- Line 289-293: The case is no longer accessible on the database, so we cannot go back and check. We think it is appropriate to leave it as a brief example.
- Line 303-321: We have provided an English explanation of tikanga.
- Line 317-318: We have clarified this slightly. "Whakapapa" can be both a noun and a verb.
- Line 405-406: We have changed the prior sentence to hopefully clarify this point. A future appeal would likely touch both the establishment of the grave risk exception and the discretion.
- Line 441-442: We thank the reviewer for their helpful thoughts on this point, and in response we have developed the point further.
Reviewer 2 Report
Overall the article is well written and interesting. It presents very up to date (even 2023) case law on recent decisions, particularly in New Zealand and Australia. I do feel that the geographical focus should somehow be reflected in the title, since the title gives the impression that courts globally might be adopting an individualised best interests approach.
Novelty: Is the question original and well-defined? Do the results provide an advancement of the current knowledge? Well this topic has been much discussed on literature in various jurisdictions, but that does not make the discussion here superfluous or less worthy, as it reflects recent trends in jurisprudence in the chosen countries.
Interest to the Readers: Are the conclusions interesting for the readership of the journal? I believe there is ongoing interest in teasing out the relationship between the best interests of the individual child and the Hauge convention’s prompt return mechanism. The section on the return of a Maori child in potential violation of her culture would also be of broader interest.
Overall Merit: Is there an overall benefit to publishing this work? Yes I believe there is merit to publishing it. I would have like to see at least a passing reference to the HCCH Guide to Good Practice on Article 13(b) of 2020.
English Level: Is the English language appropriate and understandable? Yes there are some missing references in footnotes though.
Author Response
We thank the reviewer for their helpful and thoughtful comments. We have adopted all their suggestions, and provided responses below:
- We agree that the title should be amended to clarify the geographical focus and we have done so.
- We have incorporated the Guide to Good Practice in our revised version.
- We have corrected the missing references.